# Thymine-Functionalized Gold Nanoparticles (Au NPs) for a Highly Sensitive Fiber-Optic Surface Plasmon Resonance Mercury Ion Nanosensor

**DOI:** 10.3390/nano11020397

**Published:** 2021-02-04

**Authors:** Huizhen Yuan, Guangyi Sun, Wei Peng, Wei Ji, Shuwen Chu, Qiang Liu, Yuzhang Liang

**Affiliations:** 1School of Physics, Dalian University of Technology, Dalian 116024, China; hzyuan@dlut.edu.cn (H.Y.); sunguangyi@mail.dlut.edu.cn (G.S.); g21402062@mail.dlut.edu.cn (S.C.); qiangliu@mail.dlut.edu.cn (Q.L.); yzliang@dlut.edu.cn (Y.L.); 2School of Chemical Engineering, Dalian University of Technology, Dalian 116024, China

**Keywords:** fiber-optic nanosensor, surface plasmon resonance, Au NPs, mercury ion, thymine-1-acetic acid

## Abstract

Mercury ion (Hg^2+^) is considered to be one of the most toxic heavy metal ions. Once the content of Hg^2+^ exceeds the quality standard in drinking water, the living environment and health of human beings will be threatened and destroyed. Therefore, the establishment of simple and efficient methods for Hg^2+^ ion detection has important practical significance. In this paper, we present a highly sensitive and selective fiber-optic surface plasmon resonance (SPR) Hg^2+^ ion chemical nanosensor by designing thymine (T)-modified gold nanoparticles (Au NPs/T) as the signal amplification tags. Thymine-1-acetic acid (T-COOH) was covalently coupled to the surface of 2-aminoethanethiol (AET)-modified Au NPs and Au film by 1-ethyl-3-(3-dimethylaminopropyl)carbodiimide hydrochloride/N-Hydroxysuccinimide (EDC/NHS) activation effect, respectively. In the presence of Hg^2+^ ions, the immobilized thymine combines specifically with Hg^2+^ ions, and forms an Au/thymine-Hg^2+^-thymine/Au (Au/T-Hg^2+^-T/Au) complex structure, leading to a shift in SPR wavelength due to the strong electromagnetic couple between Au NPs and Au film. Under optimal conditions, the proposed sensor was found to be highly sensitive to Hg^2+^ in the range of 80 nM–20 µM and the limit of detection (LOD) for Hg^2+^ was as low as 9.98 nM. This fiber-optic SPR sensor afforded excellent selectivity for Hg^2+^ ions against other heavy metal ions such as Fe^3+^, Cu^2+^, Ni^2+^, Ba^2+^, K^+^, Na^+^, Pb^2+^, Co^2+^, and Zn^2+^. In addition, the proposed sensor was successfully applied to Hg^2+^ assay in real environmental samples with excellent recovery. Accordingly, considering its simple advantages, this novel strategy provides a potential platform for on-site determination of Hg^2+^ ions by SPR sensor.

## 1. Introduction

Heavy metals are characterized by natural stability, biological toxicity, activity and persistency, non-biodegradation, and bioaccumulation, etc. [1,2,3,4]. Most creatures, including human beings, may be seriously poisoned if they are exposed to or ingest even small amounts of heavy metals [5,6]. Mercury (Hg) is one of the most toxic pollutants among these heavy metals [7,8]. Mercury and its derivatives, as important environmental pollutants, can spread globally via food chains and the atmosphere [9]. They can persistently exist in the environment [5,10]. Due to its acute toxicity, even a small concentration of mercury may greatly threaten animals and human beings in an irreversible way. Severe Hg poisoning may even damage the functions of internal organs of the human body, and cause neuronal destruction, cerebral atrophy, renal dysfunction, and other problems [11,12]. In addition, it is difficult for the human body to excrete mercury through metabolism owing to its accumulation in the body [13]. The World Health Organization (WHO) stipulates that Hg^2+^ ions in drinking water shall not be higher than 1 ppb [14]. Thus, it is meaningful to detect mercury ions in a fast, accurate, and quantitative way. Currently, atomic emission spectroscopy [15], atomic absorption spectrometry [16], and atomic fluorescence spectrometry [17] are mainly used for mercury ion detection. However, due to the requirement for complex pretreatment and expensive equipment, these analytical methods cannot be used for rapid on-site detection and biological toxicity studies of living organisms. Development of a small and sensitive in-situ detection scheme is urgently needed. 

As an alternative and supplement to traditional Hg^2+^ ion content analysis and detection practice, nanosensors are developing fast due to their good compatibility in analysis and detection. They have the advantages of high sensitivity, high specificity and stability, ease of operation, fast response, and so on [18,19,20]. Among them, the fiber-optic SPR sensor has the features of anti-interference, low cost, and easy integration, and it consumes few samples without the need for labeling. It has been widely used in environmental monitoring, the food industry, clinical medicine, and other fields [21,22,23]. The fiber-optic SPR sensor has already been used for Hg^2+^ detection: Raj et al. reported using a fiber-optic SPR sensor for Hg^2+^ ion detection by utilizing an Au NPs–polyvinyl alcohol (PVA) hybrid as a sensing material, however, the limit of detection (LOD) was only down to 1 × 10^−6^ M [24]. Taking advantage of a chitosan (CS)/polyacrylic acid (PAA) multilayer as the sensitive film, Chen et al. demonstrated that the detection sensitivity of a fiber-optic SPR Hg^2+^ ion sensor was up to 0.5586 nm/μM [25].

Gold nanoparticles (Au NPs) can be used to improve the sensitivity of nanosensors for small molecule detection [26,27,28,29,30] due to their quantum size effect, small size effect, surface effect, and macro quantum tunneling effect [31,32]. So far, Au NPs have been utilized to detect Hg^2+^ ions in water and soil samples through different functionalization strategies [33,34]. Au NPs as SPR signal amplification tags are mainly based on the plasmon coupling between the surface plasmons propagating on the fiber-optic SPR probe and the localized surface plasmons (LSP) oscillating around the Au NPs, which performs further signal enhancement [35]. The fiber-optic SPR sensor based on Au NPs can greatly improve its absorption intensity of the spectrum, which has broad application prospects in the field of non-destructive detection of harmful substances [36,37].

Thymine (T), as a nucleobase in nucleic acid, has been proven to be one of the most selective ligands that can specifically bind to Hg^2+^ ions between two thymine bases [38,39]. Thymine in the form of thymine-1-acetic acid (T-COOH) has been considered as an alternative receptor for detecting Hg^2+^ ions; a probe is formed by the reaction between the -COOH of thymine-1-acetic acid and the -NH_2_ [40,41]. Dai et al. prepared thymine-functionalized Ag NPs by T-COOH and AET for Hg^2+^ determination [42]. In this work, a fiber-optic SPR sensor for Hg^2+^ ion detection is proposed based on thymine (T)-modified gold nanoparticles (Au NPs/T), which were adopted as the signal amplifying tags to enhance SPR signals. For construction of such tags, thymine-1-acetic acid (T-COOH) was attached on the AET-modified Au NPs’ surface via the condensation reaction by EDC/NHS activization. Meanwhile, the same modification process was also performed on the sensing Au film surface. In the presence of Hg^2+^, a particular Au/T-Hg^2+^-T/Au NPs sandwich structure was formed, resulting in a significant shift in the SPR wavelength due to the electromagnetic coupling between Au NPs and Au film. By using this sensing principle, the sensing condition, sensitivity, and selectivity of the proposed fiber-optic SPR sensor were systemically investigated. 

## 2. Materials and Methods

1-ethyl-3-(3-dimethylaminopropyl)carbodiimide hydrochloride (EDC), chloroauric acid (HAuCl_4_·4H_2_O), thymine-1-acetic acid (T-COOH), N-Hydroxysuccinimide (NHS), and 2-aminoethanethiol (AET) were purchased from Sigma-Aldrich (Shanghai, China). Anhydrous ethanol, acetone, and 2-morpholineethanesulfonic acid (MES) were purchased from Energy Chemical (Shanghai, China). Sodium borohydride, HgCl_2_, CuCl_2_·2H_2_O, KCl, NaCl, PbCl_2_, NiCl_2_·6H_2_O, BaCl_2_, FeCl_3_·6H_2_O, ZnCl_2_, and CoCl_2_ were purchased from Sinopharm Chemical Reagent Co. (Shanghai, China). All solutions were prepared with ultrapure water (18.2 MΩ·cm) obtained from the Milli-Q purification system. 

As shown in Figure 1, the fiber-sensing part was connected with fiber-optic jumpers through two SMA905 connectors. In the detection process, the light source (Ocean Optics, HL-2000, Inc, Dunedin, FL, USA) was used to enter the fiber-optic SPR sensing area through a fiber-optic jumper, and the other end of the fiber-optic jumper was responsible for guiding the reflected light from the sensing area into an SMA fiber input interface of the spectrometer (Ocean Optics, HR4000, Inc). The optical resolution of the spectrometer was 0.25 nm and the spectral range was from 200 to 1200 nm. The SPR signal was imported into the computer through the spectrometer and the signal was analyzed and processed through the Laview program written in our group.

The sensing fiber was a multimode fiber with 400 μm core and its numerical aperture was 0.37. The fiber was cut into 8 cm pieces, and then scraped off the plastic cladding and coating layer located in the middle of the optical fiber, exposing the core part of about 5 mm long. The stripped fiber probes were washed with water, ethanol, and acetone. Subsequently, we deposited 2 nm chromium film and 50 nm Au film on the stripped fiber surface by the magnetron sputter coating apparatus, respectively. For forming AET-functionalized optic fiber, the Au-coated optic fiber was soaked in 10 mM AET solution, and left to stand at 4 °C for 6 h. Then, 0.1 mM T-COOH solution was activated with 10 mM EDC/NHS solution (pH 6.0 MES buffer solution, mixed volume ratio of 1:1), and left to stand for 15 min. The T-modified fiber-optic SPR surface was achieved by immersing the AET-functionalized optic fiber in the carboxyl-activated T-COOH solution for 3 h.

In a typical synthesis procedure of AET-functionalized Au NPs (Au NPs/AET), 290 μL HAuCl_4_·4H_2_O (1%) and 170 μL AET (213 mM) were dissolved with 79.71 mL ultrapure water, then we slowly stirred it for 20 min in darkness. Then, 4.2 μL NaBH_4_ solution (18 mM) was added into the mixture with stirring for 10 h. As shown in Figure 2, for the preparation of thymine-functionalized Au NPs (Au NPs/T), 0.1 mM T-COOH was activated by 10 mM EDC/NHS solution (MES buffer solution, 1 mM, pH 6) for 15 min at 25 °C with constant stirring. After that, 0.3 mL of carboxyl-activated T-COOH was added into 5 mL Au NPs/AET solution with stirring at room temperature for 4 h.

The standard solutions with different concentrations of Hg^2+^ ions were prepared by successively diluting the stock solution (50 μM). Hg^2+^ ion solution with different concentrations was passed into the light path for detection, and the spectrometer was used to transfer the collected data to the computer for analysis. For details, a thymine-modified fiber probe was immersed in each Hg^2+^ ion standard solution (0–20 μM) for 10 min; and then Au NPs-T tags were exposed to the sensing surface with incubation for another 30 min, leading to formation of the specific T–Hg^2+^–T complexes (Figure 1). The changes in SPR wavelength before and after the introduction of Hg^2+^ ions were recorded for sensing analysis. The detection of Hg^2+^ was repeated at least 3 times. Then, 1 mM aqueous solutions of various cations (Fe^3+^, Cu^2+^, Ni^2+^, Ba^2+^, K^+^, Na^+^, Pb^2+^, Co^2+^, and Zn^2+^) were used to test the selectivity of T-COOH-modified fiber-optic sensing system for Hg^2+^ ions. Tap water was used for demonstration of Hg^2+^ ion detection in real water samples.

## 3. Results

The sensing principle of this work was based on using thymine-functionalized Au NPs to enhance the SPR signal for detection of Hg^2+^ ions. The relatively inexpensive and chemically stable thymine-1-acetic acid was considered as an alternative receptor instead of oligonucleotides [40]. Hg^2+^ and thymine groups can form stable T-Hg^2+^-T pairs with strong affinity. If we could design thymine-functionalized Au NPs, Hg^2+^ could be activated as a bridge to link the neighboring thymine groups (T) for the formation of an Au/T-Hg^2+^-T/Au NPs sandwich structure (Figure 1). The strong electric field coupling between Au NPs and Au film is believed to be the result of a significant shift in resonance wavelength of the sensing Au film. The Hg^2+^ ion concentration-dependent wavelength shift of SPR could be utilized for the quantitative analysis of Hg^2+^ ions.

Next, we considered the construction of thymine-functionalized Au NPs and a sensing surface. AET has an amino group and a mercapto group at both ends of the alkyl chain. According to reports, the Au-S bond has a stronger force than that of the Au-N bond. The attachment of AET on Au NPs’ surface via the formation of an Au-S bond could give amino functionality to Au NPs/AET. Subsequently, amide of Au NPs/AET could react with T-COOH via condensation reaction by EDC/NHS activization, leading to the formation of Au NPs/T. The functionalization process was further characterized by XPS and UV-vis spectroscopy.

Figure 3 shows the XPS spectra with peaks of N1s, S2p, and O1s for Au NPs/AET and Au NPs/T. The peak at 399.4 eV could be related to the N–C groups of 2-aminoethanethiol being adsorbed on Au NPs’ surface (Figure 3a). The S2p zone (Figure 3b) was curve-fitted with doublet 2p1/2 and 2p3/2 signals. These two peaks, located at 161.3 and 162.6 eV, confirmed the presence of thiol groups on Au NPs/AET. For Au NPs/T, the observed XPS peak of O clearly indicated that T-COOH was attached on the surface of Au NPs/AET. Meanwhile, the asymmetric N1s peak (Figure 3d) in the spectrum of Au NPs/T comprised two peaks at 400.9 and 399.2 eV, which were ascribed to C-N and O=C-N bonding, also confirming the formation of a N-C bond via condensation reaction between the COOH group of T-COOH and the NH_2_ group of AET. It should be noted that optical fiber is constituted of silicon dioxide. For Au NPs/AET, the peak of O1s was located at 531.1 eV, which corresponded to the O atom in Si-O bond (Figure 3f). For Au NPs/T, a new O1s peak was observed at 532.7 eV, which can be ascribed to the O atom in the C-O bond. This result clearly indicates that T-COOH was attached on the surface of Au NPs/AET.

As shown in the UV–vis spectra of Figure 4a, compared with the Au NPs/AET, no obvious change in the location of the UV–vis peak of Au NPs/T was observed (UV–vis peak was 522 nm). This result indicates that the combination of Au NPs’ surface with a functional molecule did not change the stability of Au NPs. The size and morphological structure of functionalized Au NPs were observed with the JEM-2100F transmission electron microscope at 200 kV. According to Figure 4b, the synthesized Au NPs/T were a sphere with a diameter of about 25 nm, which was quite homogeneous. These results indicate the successful preparation of Au NPs/T tags, and more importantly that such tags were near monodispersed, which has great potential for SPR sensing of Hg^2+^ ions.

In order to clarify the effect of Au NPs/T tags on Hg^2+^ ions detection, the shift in resonant wavelengths of SPR during the whole detection process is shown in Figure 5. Firstly, Hg^2+^ standard solution was introduced into the sensing system, and Hg^2+^ ions were captured through forming the specific T-Hg^2+^. There was no obvious fluctuation of the SPR resonant wavelength after injecting Hg^2+^ ions due to the slight changes in the surface refractive index. Then, the nonspecific binding of Hg^2+^ ions was removed through the injection of water. Finally, Au NPs/T tags were injected into the sensing system; it was found that the SPR resonant wavelength showed a significant shift. Figure 6 shows the field emission scanning electron microscope (FE-SEM) image of the optical fiber sensing region before and after exposure to Hg^2+^ and Au NPs/T tags. As shown in Figure 6b, it distinctly indicates that an Au/T-Hg^2+^-T/Au sandwich structure was formed in the sensing region. In an Au/T-Hg^2+^-T/Au structure, the nanoparticle is coupled with Au film like its mirror image, and a strong enhanced electric field is generated between the gold film and the nanoparticle [43]. Accordingly, Au NPs/T tags can enhance the response signal of a fiber-optic SPR sensor for Hg^2+^ detection. These results clearly verified the reliability of the present SRP Hg^2+^ ion sensor.

The modification conditions were studied before application of the present sensor for the detection of Hg^2+^ ions. The concentration of T-COOH will affect the Hg^2+^ ion sensing performance of thymine-functionalization Au film on the fiber-optic surface. As shown in Figure 7, as we increased the concentration of T-COOH used in the modification process for the AET-modified fiber sensing surface, the resonant wavelength of SPR shifted toward to the long wave direction, and reached a maximum value with the T-COOH concentration of 1 mM. This result indicates that the covalent binding of T and AET approached saturation point. Therefore, 0.1 mM of T-COOH was selected as the best concentration for the functionalization of Au film located on the fiber-optic sensing surface. 

In order to find out the optimal ratio of AuNPs/AET solution and T-COOH for the preparation of Au NPs/T tags, we made carboxyl-activated 0.1 mM thymine-1-acetic acid solutions of 0.1, 0.15, 0.2, 0.3, and 0.5 mL; and a hybrid reaction with an AuNPs/AET solution of 5 mL, respectively. The reaction products were used for mercury ion detection. According to the SPR signal, 0.3 mL mixed solution of thymine-1-acetic acid and EDC/NHS showed a maximum SPR resonant wavelength shift after reacting with Au NPs/AET (Figure 8). Therefore, the optimum volume ratio is 3:50. It can be seen that compared with the Au NPs/AET, the location of the ultraviolet absorption peak of Au NPs/T showed no obvious change (Figure 4), which indicated that the combination of Au NPs surface and thymine would not change the stability of Au NPs.

Under optimized conditions for the preparations of signal amplification tags and sensing surface, the quantitative determination of Hg^2+^ ions was conducted through the SPR signal response. As shown in Figure 9a, the shift in resonance wavelength of SPR increased with the increasing Hg^2+^ ion concentration. For the sensing process, Hg^2+^ ions are just like a connection bridge which is first captured by the T on the Au film surface of fiber-optics to produce Hg^2+^-T complex, and then the captured Hg^2+^ ions combine with the T modified on the surface of the Au NPs, thus forming a stable Au/T-Hg^2+^-T/Au NPs sandwich structure, resulting in the shift of the SPR resonance wavelength. Obviously, such a sandwich structure is highly dependent on the concentration of Hg^2+^ ions. Figure 9b shows the corresponding linear region of Hg^2+^ ions’ concentration and SPR resonant wavelength; it can be seen that the SPR resonant wavelength and Hg^2+^ ions’ concentration represented a linear relation in the concentration range of 0~0.2 μM. The correlation coefficient of the linear relation was 0.985 after linear fitting. The LOD of the present sensor for Hg^2+^ ion detection was calculated to be 9.98 nM (S/N = 3), which is lower than the World Health Organization (WHO) guideline value for Hg^2+^ ions in drinking water. Table 1 shows the comparison between our proposed sensor and other SPR sensors for Hg^2+^ detection. It can be seen that the present SPR sensor is comparable to the detection limits obtained by other SPR sensors.

As this was an SPR sensing system, it was necessary to detect its selectivity. There were abundant interfering metal cations in the water samples, such as Fe^3+^, Cu^2+^, Ni^2+^, Ba^2+^, K^+^, Na^+^, Pb^2+^, Co^2+^, and Zn^2+^. For selectivity measurements, the concentrations of Hg^2+^ ions and each of the other metal cations were 20 and 1 mM, respectively. As expected, the interference from other metal cations was negligible, even when the concentrations of the other metal cations were 50 orders larger than that of Hg^2+^ ions. This result clearly demonstrates that our proposed sensor has a high selectivity for Hg^2+^ ion detection (Figure 10). Such high selectivity is due to the special combination between T and Hg^2+^ ions, which forms a relatively stable Au/T-Hg^2+^-T/Au complex structure. These results demonstrated that the present SPR sensor possesses a high potential application in environmental pollution detection.

In order to evaluate the practicability of the present fiber-optical SPR sensor for the natural samples, we extracted water samples from our laboratory faucets. The water samples were spiked with standard Hg^2+^ ions of varying concentrations (80, 100, and 200 nM) to study the recovery of Hg^2+^ ions in these samples. The experimental findings show that no Hg^2+^ ion was detected in the samples due to an extra-low concentration of Hg^2+^ ions in tap water. However, Hg^2+^ ions could be well detected in the spiked water samples (Table 2). It can be seen that recovery of Hg^2+^ ions in these samples was in the range of 95%–110%, indicating that the present fiber-optic SPR sensor possesses excellent potential for the detection of Hg^2+^ ions in real water samples.

## 4. Conclusions

We have designed thymine-functionalized Au NPs by a simple condensation reaction, which can be used as a single amplification tag for the fiber-optic SPR sensing of Hg^2+^ ions in aqueous solutions with high sensitivity and selectivity. For the proposed sensor, the resonant wavelength of SPR highly relies on the Hg^2+^ ions’ concentration-dependent sandwich structure (Au/T-Hg^2+^-T/Au), because of the electromagnetic coupling between Au NPs and Au film. Our results clearly demonstrated that the present sensor exhibits a specific response to Hg^2+^ ions. The LOD for Hg^2+^ ion detection can be down to 9.98 nM, which is much lower than the guideline value of WHO for Hg^2+^ ions in drinking water. Additionally, the present sensor exhibited an excellent recovery of Hg^2+^ ion detection in the real water samples. We believe that the present strategy is of much interest for Hg^2+^ ion analysis of on-site environmental samples.

## Figures and Tables

**Figure 1 nanomaterials-11-00397-f001:**
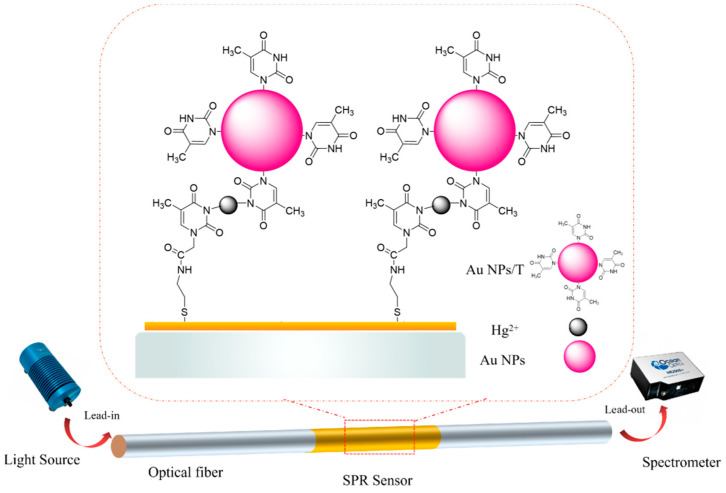
A schematic of the thymidine-functionalized fiber-optic surface plasmon resonance (SPR) biosensor for Hg^2+^ detection.

**Figure 2 nanomaterials-11-00397-f002:**
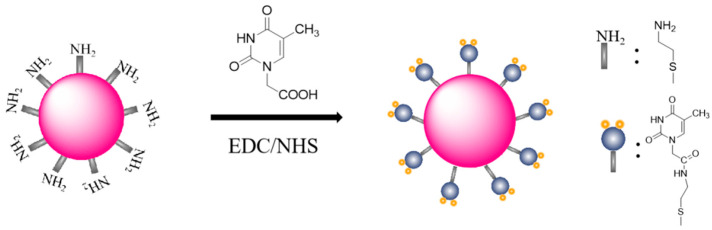
A schematic illustration of the fabrication of the thymine-modified gold nanoparticles (Au NPs/T).

**Figure 3 nanomaterials-11-00397-f003:**
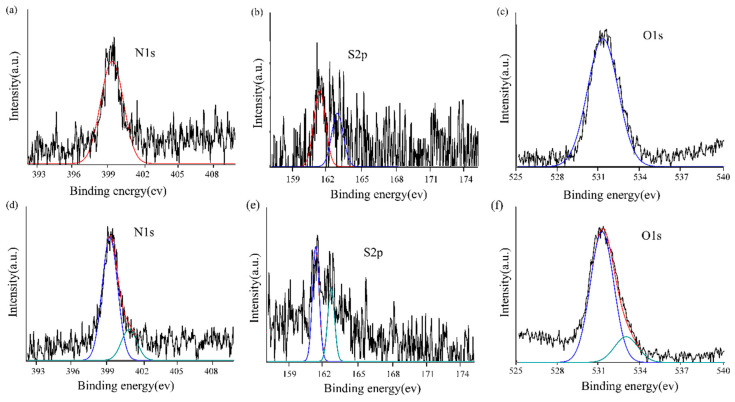
The XPS spectra of Au NPs/AET and Au NPs/T: (**a**) N1s spectrum of the Au NPs/AET; (**b**) S2p spectrum of Au NPs/AET; (**c**) O1s spectrum of the Au NPs/AET; (**d**) N1s spectrum of the Au NPs/T; (**e**) S2p spectrum of Au NPs/T; (**f**) O1s spectrum of the Au NPs/T.

**Figure 4 nanomaterials-11-00397-f004:**
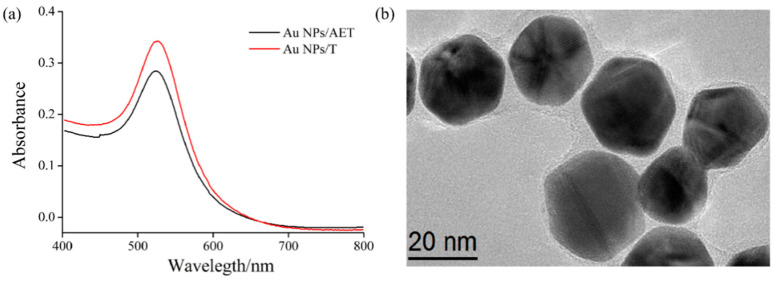
The characterization of Au NPs: (**a**) UV–vis absorption of Au NPs/AET and Au NPs/T; (**b**) TEM image of Au NPs/T.

**Figure 5 nanomaterials-11-00397-f005:**
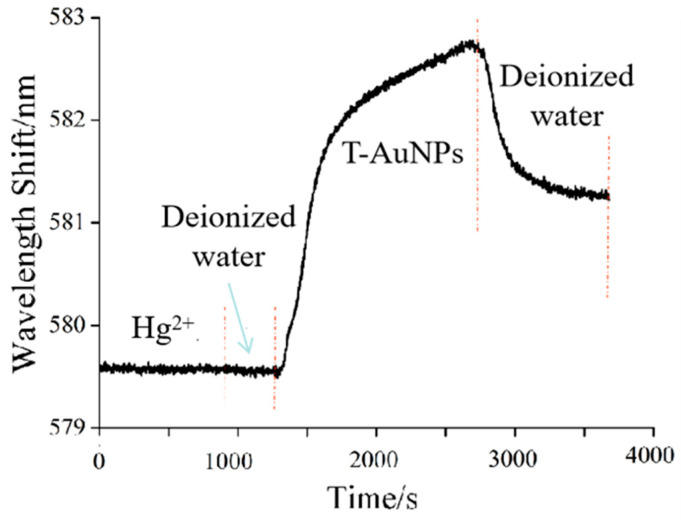
Sensing of Hg^2+^ ions with the fiber-optic SPR sensor: real time sensing process of Hg^2+^ ions.

**Figure 6 nanomaterials-11-00397-f006:**
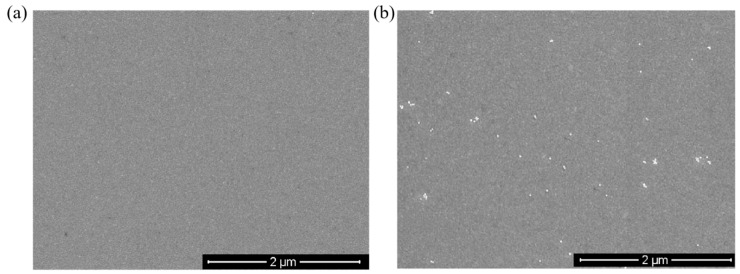
SEM image of sensing surface after the detection of Hg^2+^ ions: (**a**) without Hg^2+^; (**b**) the presence of Hg^2+^.

**Figure 7 nanomaterials-11-00397-f007:**
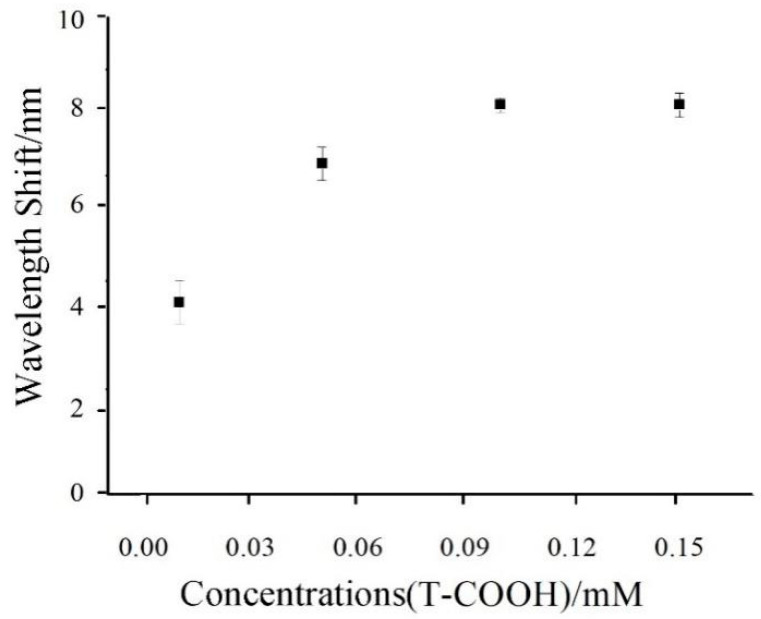
The optimization concentrations of T-COOH for the modification of a fiber-optic sensing surface.

**Figure 8 nanomaterials-11-00397-f008:**
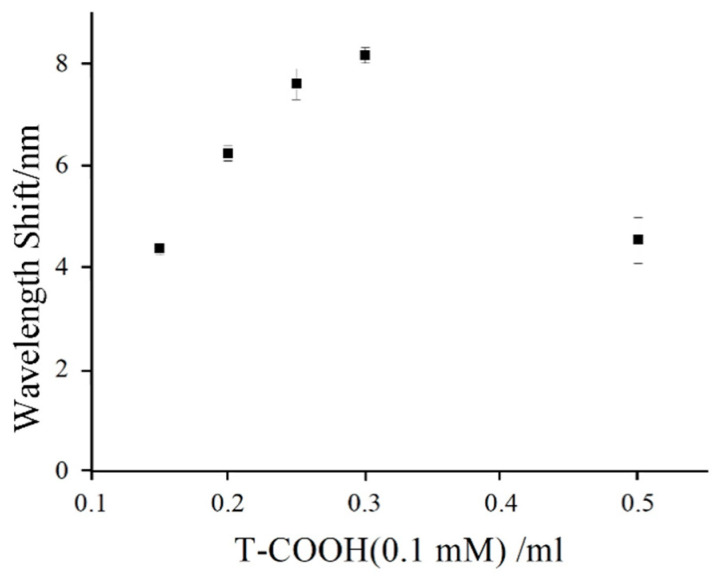
The optimization volumes of T-COOH (0.1 mM) for its modification on Au NPs.

**Figure 9 nanomaterials-11-00397-f009:**
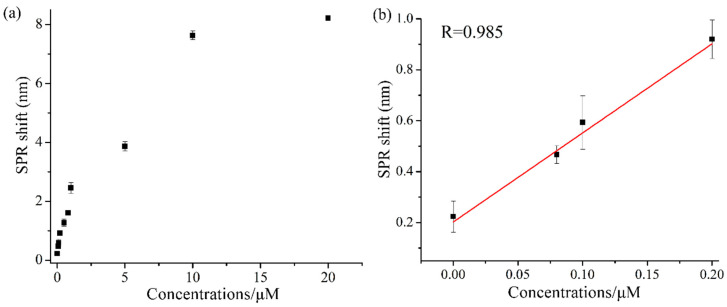
SPR signal responses of Hg^2+^ detection: (**a**) different concentrations of Hg^2+^ samples (0–20 μM); (**b**) linear range for Hg^2+^ determination at the concentrations ranging from 80 to 200 nM.

**Figure 10 nanomaterials-11-00397-f010:**
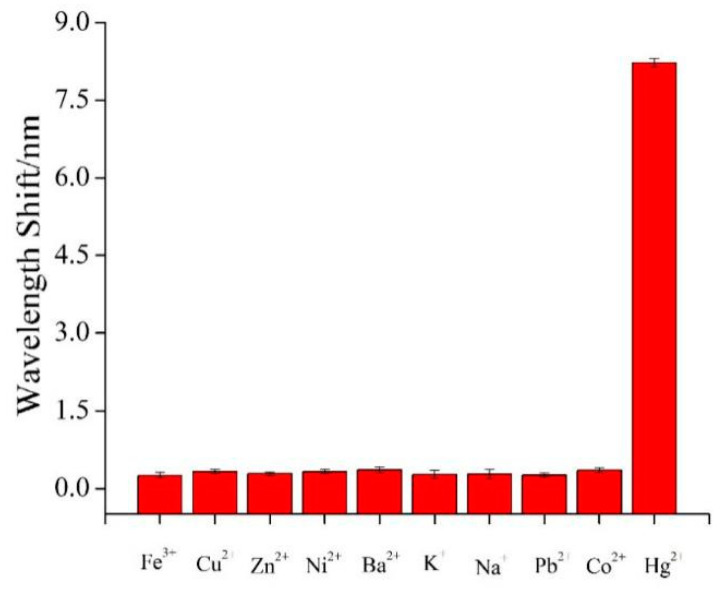
SPR responses for different metal ions based on sandwich structure.

**Table 1 nanomaterials-11-00397-t001:** Comparison of this work with some published assays for Hg^2+^ ion detection by SPR sensors.

Detection Method	Linear Range	LOD	Reference
SPR sensors	0.01–0.5 ppm	49.9 nM	[44]
Fiber-optic LSPR sensor	30 to 200 μM	30 μM	[25]
Fiber-optic LSPR sensor	1 ppb to 15 ppb	1.5 ppb	[45]
Fiber-optic SPR sensor	0.01 to 1000 µM	10 nM	[46]
SPR sensors	1–25 µM	1 µM	[24]
SPR sensors	10^1^–10^4^ µg/L	5 µg/L	[47]
SPR sensors	0.01–5 ppm	5 ppb	[48]
SPR sensors	0–100 ppb	2 ppb	[49]
Fiber-optic SPR sensor	0–20 μM	9.98 nM	This work

**Table 2 nanomaterials-11-00397-t002:** Spiked recovery of the present fiber-optic SPR sensor for Hg^2+^ ion detection in the tap water (*n* = 3).

Sample	Hg^2+^ Added /nM	Hg^2+^ Found /nM	Recovery (%) Mean ± RSD, *n* = 3
1	80	86.1 ± 7.8	108 ± 10
2	100	97.4 ± 12.6	97 ± 12.6
3	200	211.7 ± 21.3	106 ± 11

## Data Availability

Data sharing is not applicable to this article as no new data were created or analyzed in this study.

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
