# Peer review of "Thymine-Functionalized Gold Nanoparticles (Au NPs) for a Highly Sensitive Fiber-Optic Surface Plasmon Resonance Mercury Ion Nanosensor"

_nanomaterials, 2021, doi:10.3390/nano11020397_

Round 1

Reviewer 1 Report

Peng and colleagues report on thiamine-functionalized AuNPs for Hg(II) detection using SPR. The manuscript is full of basic language errors, such as ‘witch’ for ‘which’, and confusing sentences. For example, what is meant by: “They can also be transported over long distances and persistent existence”? There are unnecessary capitalizations and variations of acronyms, with 2-aminoethanethiol refered to ase both “AET” and “ATE”. The authors need to thoroughly revise the manuscript, ideally with the assistance of a native English speaker.

The notion of biodegradation can’t really be applied to metal cations – what could they possibly be degraded into? Similarly, the idea that serious poisoning occurs upon exposure to, or ingestion of, “large amounts of heavy metals” is irrelevant. Ingestion of a large amount of anything can be lethal. The point is that even small amounts of these metals can be toxic.

There is discussion in the introduction relating to AuNP use in biosensors but it has little relevance to this work as the system is not a biosensor.  

The methods are described thoroughly and the data reported are consistent with the authors’ interpretation. Figure 4a is a useful reminder that sequential surface modifications of AuNPs do not necessarily translate to changes in wavelength. The optimization is logical and rigorous. Data on selectivity and recovery of spiked samples suggests that this relatively simple system is adequate to detect mercury(II) ions from 0.2 micromolar to below the WHO guideline of 5 nm. However, a search of ‘mercury’ and ‘SPR’ generates around 150 hits. The authors must compare their range, selectivity and LOD to other methods for the readers to appreciate what makes this method superior to the others in the literature.

Author Response

  1. Peng and colleagues report on thiamine-functionalized AuNPs for Hg(II) detection using SPR. The manuscript is full of basic language errors, such as ‘witch’ for ‘which’, and confusing sentences. For example, what is meant by: “They can also be transported over long distances and persistent existence”? There are unnecessary capitalizations and variations of acronyms, with 2-aminoethanethiol refered to ase both “AET” and “ATE”. The authors need to thoroughly revise the manuscript, ideally with the assistance of a native English speaker.

Reply: Thank you for your comment. We have carefully corrected the grammatic and language errors in the revised manuscript.

  1. The notion of biodegradation can’t really be applied to metal cations – what could they possibly be degraded into? Similarly, the idea that serious poisoning occurs upon exposure to, or ingestion of, “large amounts of heavy metals” is irrelevant. Ingestion of a large amount of anything can be lethal. The point is that even small amounts of these metals can be toxic.

Reply: We have revised our description in the revised manuscript (see the last paragraph in Page 1).

Mercury ion (Hg2+) is considered to be one of the most toxic heavy metal ions.

Most creatures, including human beings, may be seriously poisoned if they are exposed to or ingested even small amounts of heavy metals

  1. There is discussion in the introduction relating to AuNP use in biosensors but it has little relevance to this work as the system is not a biosensor.

Reply: After careful consideration, biosensors in the introduction have been revised as nanosensors.

  1. The methods are described thoroughly and the data reported are consistent with the authors’ interpretation. Figure 4a is a useful reminder that sequential surface modifications of AuNPs do not necessarily translate to changes in wavelength. The optimization is logical and rigorous. Data on selectivity and recovery of spiked samples suggests that this relatively simple system is adequate to detect mercury(II) ions from 0.2 micromolar to below the WHO guideline of 5 nm. However, a search of ‘mercury’ and ‘SPR’ generates around 150 hits. The authors must compare their range, selectivity and LOD to other methods for the readers to appreciate what makes this method superior to the others in the literatur

Reply: The comparison between our proposed sensor and other SPR sensors for Hg2+ detection has been summarized in Table 1. The present SPR sensor is comparable to the detection limits obtained by other SPR sensors (see the first paragraph and Table 1 in Page 9). 

“Table 1 shows the comparison between our proposed sensor and other SPR sensors for Hg2+ detection. It can be seen that the present SPR sensor is comparable to the detection limits obtained by other SPR sensors

Table 1 Comparison of this work with some published assays for Hg2+ ion detection by SPR sensors.

Detection method

Linear range

LOD

Reference

SPR sensors

0.01–0.5 ppm

49.9 nM

[45]

Fiber-optic LSPR sensors

30 to 200 μM

30 μM

[46]

Fiber-optic LSPR sensor

1 ppb to 15 ppb

1.5 ppb

[47]

Fiber-optic SPR sensor

0.01 to 1000 µM

10 nM

[48]

SPR sensors

1–25 µM

1 µM

[49]

SPR sensors

101–104 µg/L

5 µg/L

[50]

SPR sensors

0.01–5 ppm

5 ppb

[51]

SPR sensors

0 – 100 ppb

2 ppb

[52]

Fiber-optic SPR sensor

0-20 μM

9.98 nM

This work

Ref.

45  N.I.M Fauzi, Y.W Fen, OrcID, N.A.S Omar. Silvan Saleviter,Wan Mohd Ebtisyam Mustaqim Mohd Daniyal,Hazwani Suhaila Hashim andMohd Nasrullah. Nanostructured Chitosan/Maghemite Composites Thin Film for Potential Optical Detection of Mercury Ion by Surface Plasmon Resonance Investigation. Polymers, 2020, 12, 1497.

46  Z.L Chen, K.L Han, Y.N. Zhang. Reflective Fiber Surface Plasmon Resonance Sensor for High-Sensitive Mercury Ion Detection. Appl. Sci. 2019, 9, 1480.

47  B.S. Boruah, N. Ojah, R. Biswas. Bio-Inspired Localized Surface Plasmon Resonance Enhanced Sensing of Mercury Through Green Synthesized Silver Nanoparticle. J Lightwave Technol., 2020, 38, 2086 - 2091.

48  V.P. Prakashan, G. George, M.S. Sanu, M.S. Sajna, A.C. Saritha, C. Sudarsanakumar, P.R. Biju, C. Joseph, N.V. Unnikrishnan. Investigations on SPR induced Cu@Ag core shell doped SiO2-TiO2-ZrO2 fiber optic sensor for mercury detection. Appl. Surf. Sci., 2020, 507, 144957.

49  D.R Raj, S Prasanth, T.V.Vineeshkumar, C. Sudarsanakumar. Surface Plasmon Resonance based fiber optic sensor for mercury detection using gold nanoparticles PVA hybrid. Opt. Commu., 2016, 367, 102-107.

50  Masaki Taniguchi, Mohammad Shohel Rana Siddiki, Shunsaku Ueda, Isamu Maeda. Mercury (II) sensor based on monitoring dissociation rate of the trans-acting factor MerR from cis-element by surface plasmon resonance. Biosens. Bioelectron., 2015, 67, 309-314.

51  D.D Huang, T.T Hu, N. Chen, W. Zhang, J.W Di. Development of silver/gold nanocages onto indium tin oxide glass as a reagentless plasmonic mercury sensor. Anal. Chim. Acta. 2014, 825, 51-56.

52  G.M Shukla, N. Punjabi, T. Kundu, Soumyo Mukherji. Optimization of Plasmonic U-Shaped Optical Fiber Sensor for Mercury Ions Detection Using Glucose Capped Silver Nanoparticles. IEEE Sensors Journal, 2019, 19, 3224 - 3231.

Reviewer 2 Report

The present work describes a SPR sensor for Hg ion detection. The paper is well written, detailing fabrication, measurements and protocols. Actually the structure is very similar with the group’s other works (referenced) regarding Hg sensor on very similar platform. The paper can be published if the following comments are addressed:

Explain the large concentration used for the selectivity measurements.  Also the sensor behavior (eg shift) when the concentration is increased several orders of magnitude. Maybe extend the figure 8 a graph. Looks like you work in a saturated regime.  

There are several platforms providing nM limits of detections in literature. The authors proposed one just in 2019, even with lower LOD. It is not clear to me, from the manuscript, what are the advantages of the present one.

Author Response

  1. Explain the large concentration used for the selectivity measurements. Also the sensor behavior (eg shift) when the concentration is increased several orders of magnitude. Maybe extend the figure 8 a graph. Looks like you work in a saturated regime.  

Reply: For selectivity measurements, the concentrations of Hg2+ ions and each of the other metal cations are 20 μM and 1 mM, respectively. As expected, the interference from other metal cations is negligible, even when the concentrations of the other metal cations are 50 orders larger than that of Hg2+ ions. This results clearly demonstrate that our proposed sensor has a high selectivity for Hg2+ ions detection. The shift in resonant wavelength depends on the amount of Au NPs attached on the sensing surface, and meanwhile the adsorption of Au NPs lies on the concentration of Hg2+ in the analytes. Note, two surface adsorption behaviors are involved during sensing process. Generally, the surface adsorption satisfies the Langmuir model, and thus the shift in resonant wavelength of the present sensor increases gently, when the concentration of Hg2+ ions is increased several orders of magnitude. Meanwhile, the shift in resonant wavelength represents a saturated regime.

  1. There are several platforms providing nM limits of detections in literature. The authors proposed one just in 2019, even with lower LOD. It is not clear to me, from the manuscript, what are the advantages of the present one.
    Reply: There are several platforms providing nM limits of detections in literature, but most of them are proposed by using commercial SPR devices, which limit their practical on-site applications. We have proposed a fiber-optic SPR Hg2+sensing in 2019 (ACS Sensor, 2019, 4, 704-710). While this sensor has a lower LOD than that of the present work, further research work indicates that mercaptopyridine-functionalized Au NPs do not have a long-term stability. In the present work, we utilized thymine (T) as the selective ligand for the Hg2+ Compared with previous one, no obvious change was observed in the UV-vis spectra after modification (Figure 4a), indicating that the present amplification tags (Au NPs) show a better stability than that of previous one.  

Reviewer 3 Report

In this work by Yuan et al., the authors employed thymine-functionalized Au NPs for optical detection of mercury. The results are interesting and fit the scope of Nanomaterials. However, some corrections have to be made to reach the level of discussion expected from a paper in this journal. Please see the suggestions below:
1) "Mercury ion (Hg2+) is a heavy metal ion which is difficult to be biodegraded" - the first sentence of the abstract is wrong. Ions cannot be degraded.
2) The novelty factor is not defined sufficiently well. Please specify if a similar combination of a nucleobase with nanoparticles was employed before for ion detection by anyone else. If yes, refer to these studies. If not, clearly state that this is the first report about it.
3) Fig. 1 is too small to read. Please enlarge it as the chemical formula are barely visible.
4) The level of English should be improved as there are errors e.g. "the fiber probes were putting" (Lines 110-111), "and then stir for 20 min" (Lines 120).
5) The amount of hydration water in chloroauric acid is not reported, which may make the study not reproducible.
6) O1s spectrum of Au NPs/AER should also be provided to enable comparison.
7) "Meanwhile, the asymmetric N1s peak (Figure 3(c)) in the spectrum of Au NPs/T comprises two peaks at 400.9 and 399.2 eV, which are ascribed to C-N and O=C-N bonding, also confirming that formation of N-O bond via the condensation reaction between COOH group of T-COOH and NH2 group of AET. (Figure 3(d))." (Lines 162-166). Please specify what is the reaction that you have on your mind. It is challenging to visualize what could give the N-O bond.
8) Fig. 5 - if SEM micrographs are shown. The surface before and after the process should be enclosed.
9) Why is there a shift decrease at 0.5 mL of T-COOH (0.1 mM) in Fig. 7?
10) Did the authors observed a decrease in sensorial performance upon dilution of Hg-rich water using the same specimen? Let's say that the sensor was first exposed to water having 200 nM of Hg2+. Then, the mercury content was reduced by dilution to 80 nM. Would the shift decrease or not? The latter would indicate that mercury binds irreversibly and the sensor is envisioned for single use only.

Round 2

Reviewer 3 Report

Thank you very much. I recommend the publication of the article.